# Impacts of the COVID-19 Pandemic on Food Security and Diet-Related Lifestyle Behaviors: An Analytical Study of Google Trends-Based Query Volumes

**DOI:** 10.3390/nu12103103

**Published:** 2020-10-12

**Authors:** Noor Rohmah Mayasari, Dang Khanh Ngan Ho, David J. Lundy, Anatoly V. Skalny, Alexey A. Tinkov, I-Chun Teng, Meng-Chieh Wu, Amelia Faradina, Afrah Zaki Mahdi Mohammed, Ji Min Park, Yi Jing Ngu, Sabrina Aliné, Naila Maya Shofia, Jung-Su Chang

**Affiliations:** 1School of Nutrition and Health Sciences, College of Nutrition, Taipei Medical University, Taipei 110, Taiwan; rohmah.noor29@gmail.com (N.R.M.); nganhdk91@gmail.com (D.K.N.H.); eva850602@gmail.com (I.-C.T.); ma07107007@tmu.edu.tw (M.-C.W.); amelia.faradinaa@gmail.com (A.F.); rfrf.m1999@gmail.com (A.Z.M.M.); da07107004@tmu.edu.tw (J.M.P.); ngu_yj@hotmail.com (Y.J.N.); ma07108023@tmu.edu.tw (S.A.); 2Graduate Institute of Biomedical Materials and Tissue Engineering, College of Biomedical Engineering, Taipei Medical University, Taipei 110, Taiwan; dlundy@tmu.edu.tw; 3Laboratory of Molecular Dietology, IM Sechenov First Moscow State Medical University (Sechenov University), 119146 Moscow, Russia; skalny3@microelements.ru (A.V.S.); tinkov.a.a@gmail.com (A.A.T.); 4Institute of Cellular and Intracellular Symbiosis, Russian Academy of Sciences, 460000 Orenburg, Russia; 5Department of Social and Political Sciences, Bocconi University, 20136 Milano, Italy; naila.shofia@unibocconi.it; 6Graduate Institute of Metabolism and Obesity Sciences, College of Nutrition, Taipei Medical University, Taipei 110, Taiwan; 7Nutrition Research Center, Taipei Medical University Hospital, Taipei 110, Taiwan; 8Chinese Taipei Society for the Study of Obesity (CTSSO), Taipei 110, Taiwan

**Keywords:** COVID-19, diet, nutrition, lifestyle behavior, food security, Google Trends

## Abstract

The severe acute respiratory syndrome coronavirus (SARS-CoV)-2 disease (COVID)-19 is having profound effects on the global economy and food trade. Limited data are available on how this pandemic is affecting our dietary and lifestyle-related behaviors at the global level. Google Trends was used to obtain worldwide relative search volumes (RSVs) covering a timeframe from before the COVID-19 pandemic 1 June 2019 to 27 April 2020. Spearman’s rank-order correlation coefficients were used to measure relationships between daily confirmed cases and aforementioned RSVs between 31 December 2019 and 15 April 2020. RSV curves showed increased interest in multiple keywords related to dietary and lifestyle behaviors during the COVID-19 lockdown period in March and April 2020. Spearman’s correlation analysis showed that the strongest variables in each keyword category were (1) food security (food shortage: r = 0.749, food bank: r = 0.660, and free food: r = 0.555; all *p* < 0.001), (2) dietary behaviors (delivery: r = 0.780, restaurant: r = −0.731, take-away: r = 0.731, and food-delivery: r = 0.693; all *p* < 0.001), (3) outdoor-related behaviors (resort: r = −0.922, hotel: r = −0.913, cinema: r = −0.844, park: r = −0.827, fitness: r = −0.817, gym: r = −0.811; plant: r = 0.749, sunbathing: r = 0.668, and online: r = 0.670; all *p* < 0.001), and (4) immune-related nutrients/herbs/foods (vitamin C: r = 0.802, vitamin A: r = 0.780, zinc: r = 0.781, immune: r = 0.739, vitamin E: r = 0.707, garlic: r = 0.667, omega-3 fatty acid: r = −0.633, vitamin D: r = 0.549, and turmeric: r = 0.545; all *p* < 0.001). Restricted movement has affected peoples’ dietary and lifestyle behaviors as people tend to search for immune-boosting nutrients/herbs and have replaced outdoor activities with sedentary indoor behaviors.

## 1. Introduction

One-third of the global population is currently subjected to social distancing measures to slow the spread of the severe acute respiratory syndrome coronavirus (SARS-CoV)-2 disease (COVID). The COVID-19 pandemic is not only having profound effects on healthcare systems but also on global economies, world trade, tourism, and social restrictions. These restrictions are directly impacting mental health [1,2], food security [3], food waste [4], purchasing behaviors [5], and physical activities [6].

A joint statement on COVID-19 impacts on food security and nutrition was recently released by the Food and Agriculture Organization of the United Nations (FAO), the International Fund for Agricultural Development (IFAD), the World Bank, and the World Food Program (WFP) on the occasion of the Extraordinary G20 Agriculture Minister’s Meeting, which concluded that the “pandemic is already affecting the entire food system and collective action is needed to ensure that markets are well-functioning” [7]. In addition, a joint statement on nutrition in the context of the COVID-19 pandemic in Asia and the Pacific by the FAO, WFP, World Health Organization (WHO), and United Nations Children’s Fund (UNICEF), also emphasized the importance of healthy diets, micronutrient supplementation, and nutrition surveillance especially among those most affected, such as the poor and physically vulnerable [8]. Maintaining healthy dietary and lifestyle behaviors during the COVID-19 pandemic is important for combating viral infections and maintaining mental health and well-being [9].

Globally, an estimated 3.4 billion people have access to the internet, and online information has grown in popularity since 1990 [10]. The accessibility of the internet and the rise of social media have affected our social lives and also our dietary and lifestyle behaviors [11,12]. The internet provides immediate access to an enormous amount of information, and infodemiology has been used to assess human behaviors related to the COVID-19 pandemic [13]. Google Trends is the most popular tool to gather information on web-based behaviors, and it can be used to predict or prevent health-related issues [14]. Currently, limited data are available as to how the COVID-19 pandemic is affecting our dietary and lifestyle-related behaviors at the global level. To address this shortcoming, we used Google Trends to analyze relevant keywords related to these topics.

## 2. Materials and Methods

### 2.1. Confirmed COVID-19 Cases

Data of the geographic distribution of the COVID-19 cases between 31 December 2019 and 25 April 2020 were obtained from the European Centre for Disease Prevention and Control (https://www.ecdc.europa.eu), which is a collection of COVID-19 data maintained by Our World in Data (https://ourworldindata.org/coronavirus).

### 2.2. Data Acquisition

Google Trends (https://trends.google.com.tw/trends/?geo=TW) was used to obtain worldwide relative search volumes (RSVs) covering a timeframe from before the COVID-19 pandemic (1 June 2019 to 27 April 2020) and the data download was on 30 April 2020 covering a time period of 1 June 2019 to 27 April 2020. The search period covered before and during the outbreak of the COVID-19 pandemic to reflect changes in relative interests in “coronavirus” and “dietary and lifestyle behaviors” related keywords. An RSV represents the relative search frequency and is reported on a scale of 0 to 100. Each data point represents the proportion of a search key word (as a search term) of total searches in a given country or worldwide during the time period selected. Up to a maximum of five search keywords are allowed in Google Trends, which enables comparisons of more than one search term. A value of the RSV of 100 indicates peak popularity in a given period and location and corresponds to the highest popularity if multiple search keys are used. An RSV of <1% indicates no or a very low search volume in a given country or time.

### 2.3. Search Terms

We first searched for user-specified terms related to the coronavirus among all searches performed using Google Trends. Prior to identifying the included search terms in each category, we searched several terms in Google Trends platform to understand the global search interests related to changes of dietary behavior and lifestyle during the COVID-19 pandemic which can be seen in the Appendix A. We selected those search terms that showed significant trend which were considered, including “coronavirus”, “Covid-19”, “Covid 19”, “Covid”, and “SARS-CoV2” search terms. With the goal of identifying searches most likely to be related to dietary and lifestyle behaviors during the COVID-19 outbreak, multiple keywords in four categories were analyzed: (1) food security (e.g., food bank, free food, and food shortage), (2) dietary behaviors (e.g., restaurant, food delivery, and take away/take out), (3) immune-related nutrients/herbs (e.g., vitamin A/B/C/D/E, zinc, iron, selenium, omega 3, herb, turmeric, garlic, ginger, and onion), and (4) outdoor/indoor lifestyles and behaviors (e.g., Netflix, Nintendo, recipe, cinema, hotel, resort, park, gym, exercise, yoga, cycling, aerobics, sunbathing, and outdoor). To improve the quality and reproducibility of the study, we followed Nuti et al.’s [15] checklist for documentation of Google Trends searches (Appendix A).

### 2.4. Statistical Analysis

Statistical analyses were conducted using GraphPad Prism 5 (La Jolla, CA, USA) and R Studio (vers. 1.0, R Studio, Boston, MA, USA). We descriptively analyzed changes in web search queries related to dietary and lifestyle behaviors of RSVs (as search terms) at the global level in the half year before the COVID-19 outbreak occurred. Normality was assessed by the Kolmogorov-Smirnov test. Spearman’s rank-order correlation coefficient was used to measure relationships between the aforementioned RSVs and coronavirus (as daily confirmed cases, cumulative cases, and “coronavirus RSV”) between 31 December 2019 and 25 April 2020. Interpretation of the correlation, r, was categorized as follows: r ≤ 0.2~0.1 very weak, r ≥ 0.3~0.5 fair, r ≥ 0.6~0.7 moderate, r ≥ 0.8~0.9 very strong, and r = 1 perfect [16]. A correlation-based network map was plotted using R Studio (vers. 1.0, R Studio, Boston, MA, USA).

## 3. Results

### 3.1. COVID-19

On 31 December 2019, health authorities in China reported the first case of COVID-19 to the WHO (Figure 1A). Confirmed cases sharply increased after 15 March 2020, which was followed by an announcement of the COVID-19 outbreak as a pandemic by the WHO on 12 March 2020. Of five keywords used to search for the coronavirus, namely “COVID-19”, “COVID 19”, “COVID”, “Coronavirus” and “SARS-CoV2”, “Coronavirus” yielded the greatest search volumes, with the first peak volume starting on 16 January 2020 and the second peak on 15 March 2020 (Figure 1A).

### 3.2. Food Security

RSV curves showed increased interest in keywords related to food security (“food bank/free food/free meal/food shortage”) during the COVID-19 lockdown period in March and April 2020 (Figure 1B). The top five search countries ranged across all continents from developed (e.g., the USA, Canada, the UK, Germany, Republic of Ireland, Australia, New Zeeland, Singapore) to developing countries (e.g., India, Indonesia, the Philippines, Vietnam, Egypt, United Arab Emirates, Nigeria, and South Africa) (Figure 1C–F).

### 3.3. Diet-Related Lifestyle Behaviors

We next investigated search interest related to dietary and lifestyle behaviors during the COVID-19 pandemic. Figure 2A shows that the global interest in “restaurant” substantially decreased, while “delivery” and “food delivery” increased. Sharp increases of interest in indoor behavior-related terms (e.g., Netflix, Nintendo, recipe, and cake) (Figure 2B) were seen, along with decreases in outdoor-related terms (e.g., hotel, cinema, park, and resort) observed during the COVID-19 lockdown period in March and April 2020 (Figure 2C). Although interest in “gym” decreased, interest in “exercise”, “outdoor”, and “plant” increased in March and April 2020 (Figure 2D). However, the top five search countries ranged across all continents from developed to developing countries (Appendix A).

### 3.4. Immune-Related Nutrients and Herbs

RSV curves showed sharply increased interest in keywords related to vitamins particularly vitamin C (Figure 3A) and to a lesser extent, herbs and turmeric (Figure 3B) during the COVID-19 pandemic period. The worldwide search map shows that Asian, Middle Eastern, and African countries were more likely to search for “vitamin C and coronavirus” (Figure 3C), European countries for “vitamin D and coronavirus” (Figure 3D), and Central/South American and Caribbean countries for “zinc and coronavirus” (Figure 3E). In addition, Caribbean (e.g., Jamaica) and African countries tended to search for “garlic and coronavirus” (Figure 3F) or “turmeric and coronavirus” (Figure 3G) or “herb” (Figure 3H).

### 3.5. Correlations between Dietary and Lifestyle Behavior-Related RSVs and Confirmed COVID-19 Cases

The correlation network map showed that daily confirmed COVID-19 cases were significantly positively correlated with search keywords related to food security, dietary behaviors, immune-related nutrients/herbs/food, and indoor lifestyle behaviors (e.g., plant, flower, and meditation) and inversely correlated with outdoor activities (Figure 4).

Spearman’s correlation analysis showed that the strongest variables in each keyword category were (1) food security (food shortage: r = 0.749, food bank: r = 0.660, and free food: r = 0.555; all *p* < 0.001), (2) dietary behaviors (delivery: r = 0.780, restaurant: r = −0.731, take away, r = 0.731, and food delivery: r = 0.693; all *p* < 0.001), (3) outdoor−related behaviors (resort: r = −0.922, hotel: r = −0.913, cinema: r = −0.844, park: r = −0.827, fitness: r = −0.817, gym: r = −0.811; plant: r = 0.749, sunbathing: r = 0.668, and online: r = 0.670; all *p* < 0.001), and (4) immune−related nutrients/herbs/foods (vitamin C: r = 0.802, vitamin A: r = 0.780, zinc: r = 0.781, immune: r = 0.739, vitamin E: r = 0.707, garlic: r = 0.667, omega-3 fatty acid: r = −0.633, vitamin D: r = 0.549, and turmeric: r = 0.545; all *p* < 0.001) (Table 1: COVID-19 confirmed cases). Similar correlation patterns were observed for COVID-19 cumulative cases or “coronavirus RSV” (Table 1).

## 4. Discussion

The COVID-19 pandemic has affected billions of people, who are either being killed, infected, or remain healthy. Our study found that during the COVID-19 pandemic, people were highly concerned about food security and immune-boosting interventions. This suggests that the lockdown has greatly affected the economy, food industry, and behavioral/nutritional choices. Restricted movement has also affected peoples’ dietary and lifestyle behaviors, as people tend to replace outdoor activities with sedentary indoor behaviors. For example, people are more likely to use online shopping, cook at home, use food-delivery services, watch Netflix, or play video games. Social distancing is also stressful, and people seem to seek refuge in nature as searches for “outdoor”, “plant”, “flower”, “sunbathing”, “cycling”, and “meditation” were significantly positively correlated with daily confirmed cases of COVID-19.

We have observed worldwide concerns with immune-boosting nutrients/herbs during the COVID-19 pandemic. Spearman’s correlation analysis showed strong to moderate correlations between COVID-19 confirmed cases and vitamin C (r = 0.802), vitamin A (r = 0.780), zinc (r = 0.781), vitamin E (r = 0.707), and to a lesser extent, garlic (r = 0.667), omega-3 fatty acid (r = −0.633), vitamin D (r = 0.549), and turmeric (r = 0.545). An adequate and balanced diet provides sufficient nutrients to support a healthy immune system against respiratory tract infections such as coronavirus infection. Vitamins and minerals play vital roles in energy metabolism and maintaining an effective immunological defense system. Obesity is regarded as a preexisting disease associated with COVID-19 mortality [17], disease complications, and severe symptoms [18,19]. Obesity is a state of low-grade inflammation, and coronavirus infection may further increase pro-inflammatory cytokine release and oxidative stress leading to exacerbation of a “cytokine storm” [20]. Vitamins (A, C, D, and E) exert anti-inflammatory or antioxidative effects, which may prevent a virus-induced cytokine storm and prevent tissue damage. High doses of vitamin C supplementation have been used to treat the common cold and infectious diseases, and Cheng et al. speculated that early and high intravenous dose of vitamin C injection may prevent and treat coronavirus infection [21]. Grant et al. also proposed that vitamin D supplementation may reduce risk of coronavirus infection through regulating macrophage host defense system to decrease viral replication rate [22]. Vitamin D may reduce the risk of coronavirus infection through regulating the macrophage host defense system to decrease the viral replication rate. Vitamin D is an important anti-inflammatory nutrient. However, patients with obesity had decreased levels of serum vitamin D compared to normal weight [23] and vitamin D supplementation may reduce the risk of virus induced cytokine storm in patients with obesity or chronic diseases [22]. Zinc is an essential trace element that is important for cellular functions. Zinc may reduce the viral replication rate via inhibiting SARS-CoV RNA polymerase or increase synthesis of the antioxidative enzyme, superoxide dismutase [24]. Herbal plants like turmeric may also exert immune-boosting effects or antiviral functions. However, WHO has warned against the use of traditional herbs or traditional remedies as treatment method for COVID-19 infection due to the lack of evidence-based study. Overall, future studies are needed to clarify protective roles of nutrients/herbs against infectious disease like SARS-CoV2 infection.

Our results are in agreement with a recent study by Ammar and colleagues who found that lifestyle behaviors dramatically changed during the COVID-19 pandemic period [6]. Ammar et al. conducted an international online survey in April 2020 with 1047 participants to investigate the effects of COVID-19 restrictions on physical activities and food behavior across continents ranging from Europe, North Africa, Western Asia and North America, and authors reported that COVID-19 restrictions had negative effects on physical activities (vigorous intensity, moderate intensity, walking, and all physical activities) but positive effects on daily sitting time (which increased 28%) [6]. In addition, home confinement was also associated with unhealthy dietary patterns as participants reported increased frequencies of eating unhealthy food, eating out of control, snacking between meals, and having an increased number of meals per day [6,25]. In contrast, a study of dietary behaviors of the Spanish adult population resulted in outlined healthier dietary behaviors (e.g., decreased the intake of fried foods, snacks, fast foods, red meat, pastries, or sweet beverages, but increased olive oil, vegetables, fruits or legumes) during the confinement during the COVID-19 outbreak when compared to previous habits [26]. Our global RSV data found that people were more likely to watch Netflix, play video games, cook at home, and use food delivery services following the government’s orders for home confinement or social distancing. Restrictions also sharply decreased outdoor activities (e.g., restaurants, cinemas, hotels, resorts, parks, and gyms). It is well known that sedentary indoor behaviors are associated with unhealthy dietary patterns. For example, Jezewska-Zychowicz et al. showed that “meat and meat products” were positively associated with television viewing [27]. Santaliestra-Pasías and colleagues also reported that television viewing or using the internet for recreational reasons was positively associated with “snacking behavior” but negatively correlated with “plant-based” and “breakfast” dietary patterns in European adolescents [28].

The current study found that during the COVID-19 pandemic, people were highly concerned about food security, and these concerns were shared across all continents in both developed and developing countries. Our RSV results showed that in countries (e.g., the USA, UK, and New Zealand) with existing charitable food organizations like “food banks”, those organizations had increased search interest during the COVID-19 pandemic. In contrast, developing countries, like the Philippines, Indonesia, Pakistan, and the UAE, which were less equipped with national food charity organizations tended to search for “free food”. The current study supports recent joint statements announced by the FAO, WFP, WHO, and UNICEF which emphasize that “access to food is negatively affected by income reduction, unemployment, and restriction in global food trade and government should implement scheme supporting access to food for the poor and the disadvantaged as well as those whose income is most affected” [7]. Food insecurity may cause malnutrition and affect human health. Hence, the Asia United Nations Network recommended that a prioritized set of actions and policy guidance should be given to support nutrition, particularly for those vulnerable and poor [8].

Limitations of this study include the use of English keywords and Google as the search engine, which did not capture true global interests. Google Trends analysis reflects interests, and RSV data were only relative volumes and not absolute values. The current study was also limited by its retrospective nature, and additional surveillance is needed to confirm changes in behaviors as well as health outcomes related to these changes during the COVID-19 pandemic. The strengths of the study included use of worldwide RSV data and comprehensive keyword searches related to food security, dietary behaviors, indoor and outdoor lifestyle behaviors, and immune-regulating nutrients and herbs. Spearmen’s correlation coefficient analysis showed similar correlation strengths of “food and lifestyle RSV” with “COVID-19 daily confirmed cases”, “COVID-19 cumulative cases”, and “coronavirus RSV”, suggesting similar predictive effects between confirmed cases and coronavirus RSVs. Last but not least, the monitoring of online search queries provides insights into worldwide human behavioral changes in response to COVID-19 lockdowns and social distancing.

## 5. Conclusions

In conclusion, this study provides insights into human behaviors during the 2020 COVID-19 pandemic. Specifically, our study showed profound effects of COVID-19 on sedentary indoor behaviors and global concerns with immune-boosting nutrients/herbs, and food security. Swift action is needed to strengthen the resilience of food systems, especially targeting those most vulnerable groups and food-insecure regions. Authors should discuss the results and how they can be interpreted from the perspective of previous studies and of the working hypotheses. The findings and their implications should be discussed in the broadest context possible. Future research directions may also be highlighted.

## Figures and Tables

**Figure 1 nutrients-12-03103-f001:**
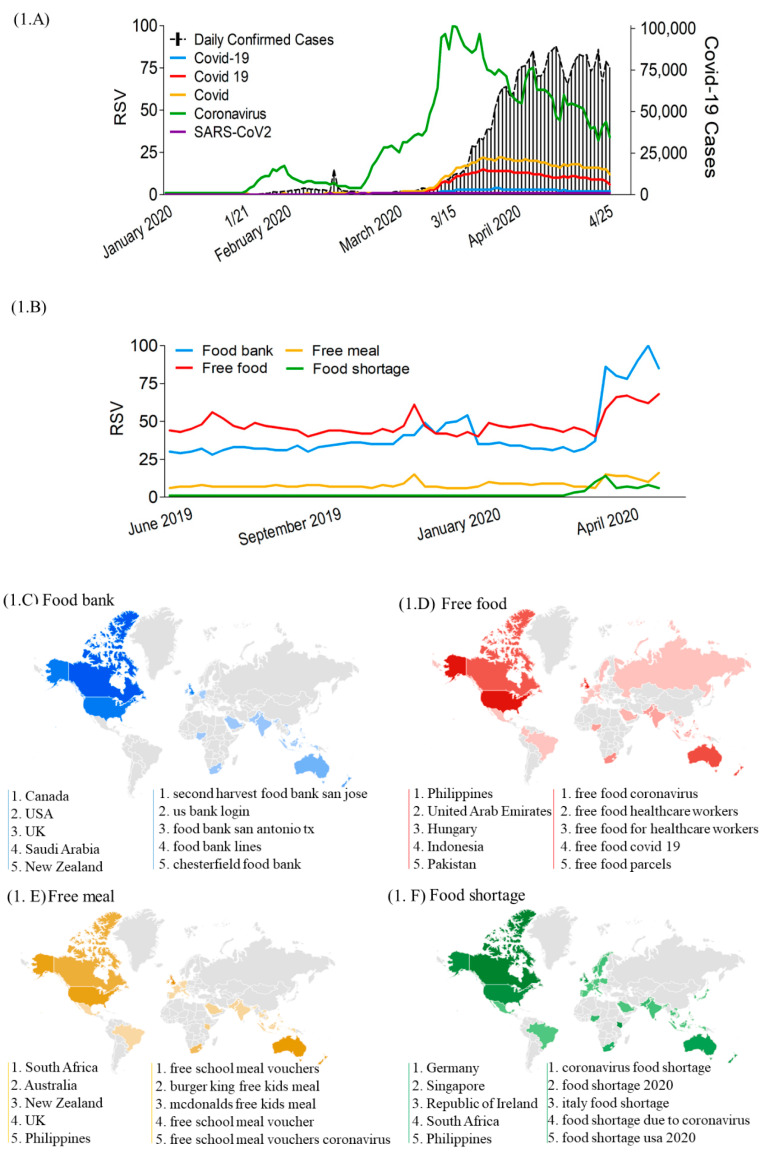
Global trend curves and heat map for nutrient and herb-related search terms. (**A**) Global trend curves for coronavirus-related search terms, cumulative confirmed coronavirus cases, (**B**) food insecurity RSV curves. (**C**–**F**) Indicates food insecurity interest heat map covering from 31 December 2019 to 25 April 2020: (**C**) Food bank, (**D**) Free food, (**E**) Free meal, (**F**) Food shortage. (**C**–**F**) The left line indicates the top five searched countries and the right line indicated the top five rising queries related search term. RSV: relative search volumes.

**Figure 2 nutrients-12-03103-f002:**
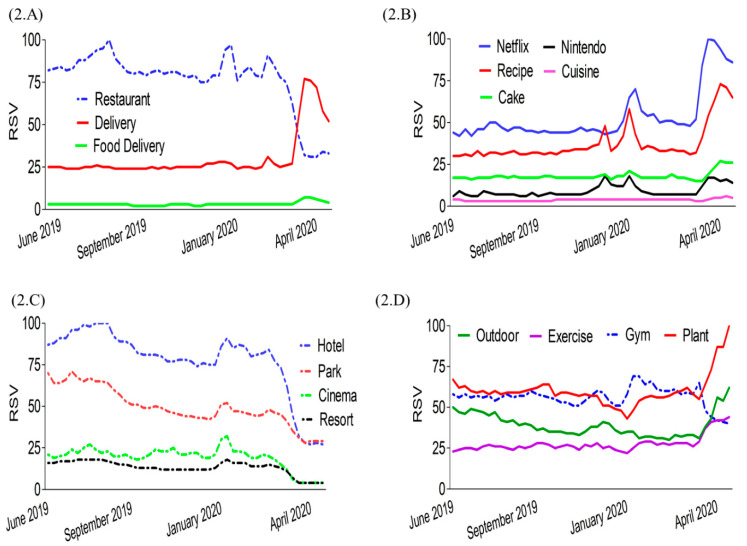
Global trend curves for dietary and lifestyle behavior-related search terms from 1 June 2019 to 27 April 2020. (**A**) Dietary behavior, (**B**) Lifestyle: indoor behavior, (**C**,**D**) Lifestyle: outdoor behavior. RSV: relative search volumes.

**Figure 3 nutrients-12-03103-f003:**
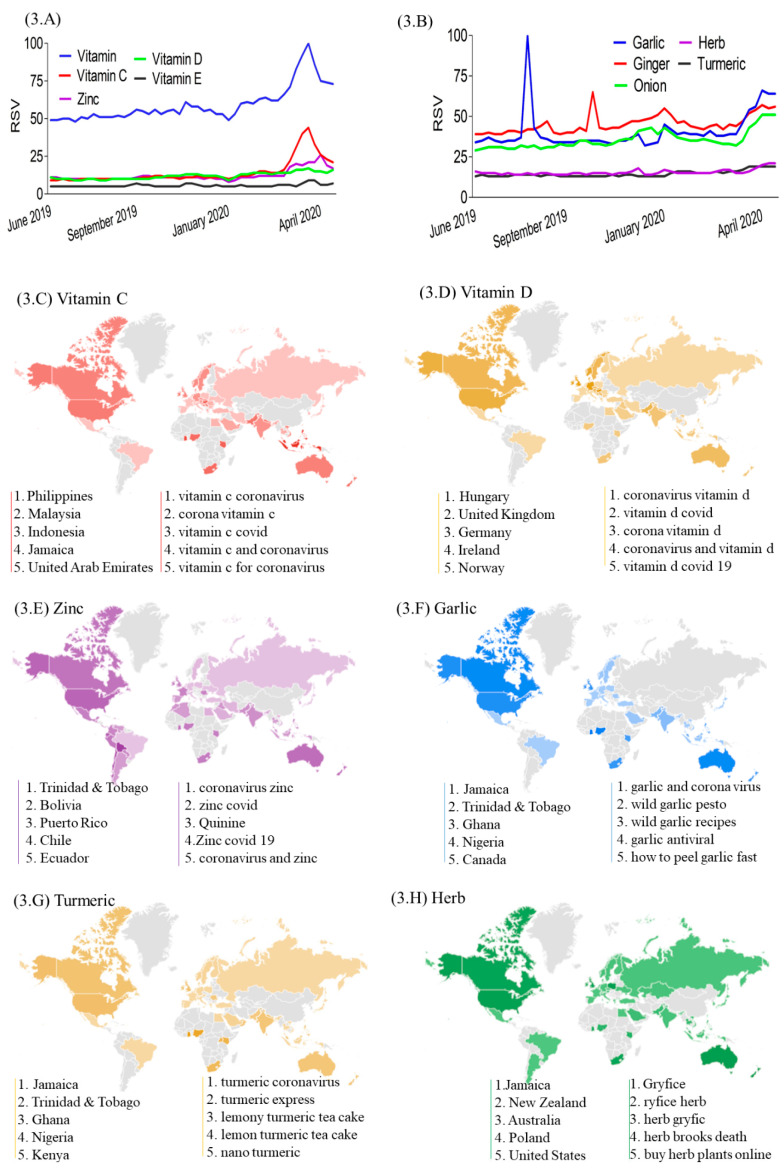
Global trend curves and heat map for nutrient and herb-related search terms. (**A**, **B**) RSV curves for nutrient and herb-related search terms. (**C**–**H**) Indicates nutrient and herb-related search terms interest heat map covering from 31 December 2019 to 25 April 2020: (**C**) Vitamin C, (**D**) Vitamin D, (**E**) Zinc, (**F**) Garlic, (**G**) Turmeric, (**H**) Herb. (**C**–**H**) The left line indicates the top five searched countries and the right line indicated the top five rising queries related search term. RSV: relative search volumes.

**Figure 4 nutrients-12-03103-f004:**
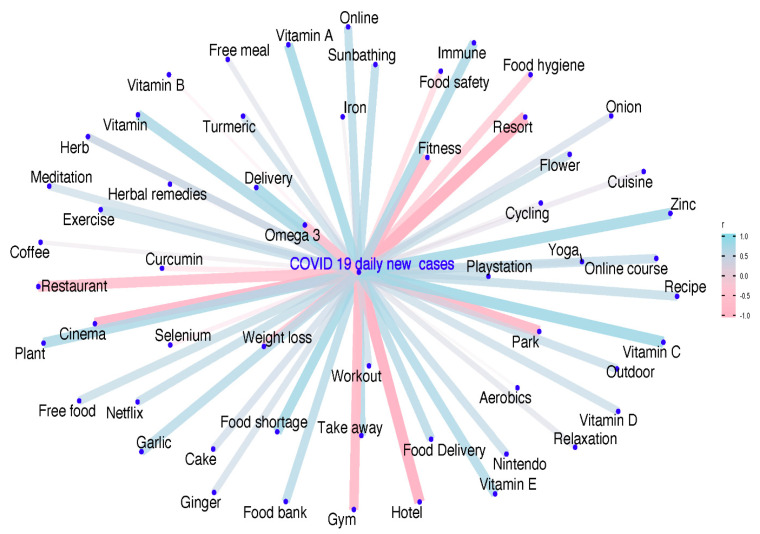
Global network correlations between daily conformed COVID-19 cases and diet-related lifestyle behavior search terms. The size of the nodes represents the strength of the correlation between the daily confirmed COVID-19 cases and diet-related lifestyle behavior search terms denoted as relative search volumes (RSVs). Each path represents a correlation, and the wider and less transparent the path, the stronger the correlation between the two variables. Blue indicates a positive correlation, while red represents a negative correlation.

**Table 1 nutrients-12-03103-t001:** Spearman’s correlation coefficients of dietary and lifestyle behavior-related search terms and daily confirmed COVID-19 cases, cumulative COVID-19 cases, and coronavirus Google Trends search terms, 31 December 2019 to 25 April 2020.

Search Query	Spearman’s Correlation					
	COVID-19 Daily Confirmed Cases		COVID-19 Cumulative Confirmed Cases		“Coronavirus” Google Trend Search Volume	
	r	*p* Value	r	*p* Value	r	*p* Value
Food security						
Food bank	0.679	<0.001	0.674	<0.001	0.625	<0.001
Free food	0.583	<0.001	0.564	<0.001	0.437	<0.001
Free meal	0.347	<0.001	0.327	0.000	0.183	0.048
Food shortage	0.768	<0.001	0.828	<0.001	0.885	<0.001
Dietary behavior						
Restaurant	−0.733	<0.001	−0.776	<0.001	−0.744	<0.001
Delivery	0.770	<0.001	0.714	<0.001	0.642	<0.001
Food delivery	0.686	<0.001	0.662	<0.001	0.602	<0.001
Take away	0.726	<0.001	0.727	<0.001	0.639	<0.001
Lifestyle: indoor behavior						
Recipe	0.591	<0.001	0.573	<0.001	0.398	<0.001
Cuisine	0.314	0.001	0.303	0.001	−0.032	0.734
Cake	0.514	<0.001	0.478	<0.001	0.169	0.068
Netflix	0.585	<0.001	0.553	<0.001	0.504	<0.001
Nintendo	0.570	<0.001	0.567	<0.001	0.481	<0.001
Lifestyle: outdoor behavior						
Cinema	−0.844	<0.001	−0.883	<0.001	−0.780	<0.001
Hotel	−0.913	<0.001	−0.943	<0.001	−0.864	<0.001
Resort	−0.922	<0.001	−0.937	<0.001	−0.878	<0.001
Park	−0.827	<0.001	−0.820	<0.001	−0.768	<0.001
Gym	−0.811	<0.001	−0.832	<0.001	−0.610	<0.001
Exercise	0.599	<0.001	0.622	<0.001	0.376	<0.001
Outdoor	0.575	<0.001	0.611	<0.001	0.355	<0.001
Workout	0.540	<0.001	0.551	<0.001	0.394	<0.001
Yoga	0.309	0.001	0.289	0.002	0.151	0.105
Sunbathing	0.668	<0.001	0.649	<0.001	0.513	<0.001
Cycling	0.194	0.036	0.255	0.006	−0.095	0.310
Fitness	−0.817	<0.001	−0.838	<0.001	−0.608	<0.001
Aerobics	0.050	0.592	0.038	0.688	−0.174	0.060
Plant	0.749	<0.001	0.844	<0.001	0.567	<0.001
Flower	0.581	<0.001	0.567	<0.001	0.185	0.045
Immune-related nutrients/herbs						
Vitamins	0.752	<0.001	0.800	<0.001	0.913	<0.001
Vitamin A	0.780	<0.001	0.813	<0.001	0.741	<0.001
Vitamin B	0.707	<0.001	0.745	<0.001	0.644	<0.001
Vitamin C	0.802	<0.001	0.827	<0.001	0.957	<0.001
Vitamin D	0.549	<0.001	0.594	<0.001	0.705	<0.001
Vitamin E	−0.102	0.276	−0.004	0.966	−0.044	0.637
Zinc	0.781	<0.001	0.817	<0.001	0.857	<0.001
Iron	0.175	0.060	0.198	0.032	−0.110	0.239
Selenium	−0.205	0.027	−0.162	0.081	−0.194	0.036
Omega 3	−0.633	<0.001	−0.598	<0.001	−0.594	<0.001
Turmeric	0.545	<0.001	0.573	<0.001	0.409	<0.001
Garlic	0.667	<0.001	0.654	<0.001	0.472	<0.001
Ginger	0.484	<0.001	0.487	<0.001	0.350	<0.001
Onion	0.471	<0.001	0.435	<0.001	0.209	0.024
Herbs	0.480	<0.001	0.537	<0.001	0.264	0.004

## Data Availability

The datasets used and/or analyzed during the current study are available from the corresponding author on reasonable request.

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
