# Peer review of "Impacts of the COVID-19 Pandemic on Food Security and Diet-Related Lifestyle Behaviors: An Analytical Study of Google Trends-Based Query Volumes"

_nutrients, 2020, doi:10.3390/nu12103103_

Round 1
Reviewer 1 Report
Thank you for allowing me to review this contribution entitled
“Impacts of the COVID-19 pandemic on food security and diet-related lifestyle behaviors: an analytical study of Google Trends-based query volumes”
In general, the topic of research and the approach of the study are interesting.
There is consistency among research title, the research question/goal and contribution/answers for research questions but I suggest some changes to incorporate in the document, specially for a greater interest in the area of dietary habits / behaviors
Introduction
L67 It would be interesting to add a reference on how the internet and social networks influence dietary and healthy habits.
L74 to L100 are repeated
Materials and methods
L120 It would be convenient to include if the search and download date coincide, as suggested by Nuti et al and the authors referenced
L124 It is necessary to better justify the keywords used in each category. Specially in those related to Dietary behavior
Results
L 158 It’s not mentioned if there are differences between countries for the Diet-related lifestyle, as it appears in the rest of the categories.
Discussion
L212 The authors refer to “ For those who remain relatively healthy,..” but I understand that the health status of those who have carried out the searches is not known.
In general, it is surprising that the authors did not use search terms related to food, food patterns or type of diets (plant-based, weight loss diets, ...) distribution of meals, in addition to food delivery, shopping online, etc. They mention (L250) studies where questions of this type are addressed to know changes in dietary behaviors (Aman et al).
Likewise, the limitation in language, as they indicate in L281, is especially important when comparing results by countries.
Bibliography
Check the correct citation of reference 6.
Author Response
Reviewer 1 comments:
Comments to the Author:
Thank you for allowing me to review this contribution entitled “Impacts of the COVID-19 pandemic on food security and diet-related lifestyle behaviors: an analytical study of Google Trends-based query volumes”. In general, the topic of research and the approach of the study are interesting. There is consistency among research title, the research question/goal and contribution/answers for research questions but I suggest some changes to incorporate in the document, specially for a greater interest in the area of dietary habits / behaviors.
Response: Thank you for your positive response to our work and we have address those comments accordingly.
Introduction:
Point 1. L67 It would be interesting to add a reference on how the internet and social networks influence dietary and healthy habits.
Response 1: Thank you for your suggestion. we have added two new references for line 67 sentences as below:
” The accessibility of the internet and the rise of social media have affected our social lives and also our dietary and lifestyle behaviors”. (Page 2, Line 67-68).
- Pollard, C.M.; Pulker, C.E.; Meng, X.; Kerr, D.A.; Scott, J.A.J.J.o.m.I.r. Who uses the Internet as a source of nutrition and dietary information? An Australian population perspective. 2015, 17, e209.
- McGloin, A.F.; Eslami, S.J.P.o.t.N.S. Digital and social media opportunities for dietary behaviour change. 2015, 74, 139-148.
Point 2. L74 to L100 are repeated
Response 2: Sorry for the careless mistake we have made. We deleted repeated paragraphs in introduction section (Page 2, Line 74-100)
Materials and methods
Point 3.L120 It would be convenient to include if the search and download date coincide, as suggested by Nuti et al and the authors referenced.
Response 3: Thank you for your comment. we have added this information in Materials and Methods section: 2.2 Data acquisition as “Google Trends (https://trends.google.com.tw/trends/?geo=TW) was used to obtain worldwide relative search volumes (RSVs) covering a timeframe from before the COVID-19 pandemic (1 June 2019 to 27 April 2020) and the data download was on 31 April 2020 covering a time period of 1 June 2019 to 27 April 2020.” (Page 2, Line 82-85).
Point 4. L124 It is necessary to better justify the keywords used in each category. Specially in those related to Dietary behavior
Response 4: Thank you for your comment. Prior to identify the included keyword in each category, we have searched several keywords in Google Trends platform to understand what is the global search interests related to changes of dietary behavior and lifestyle during COVID-19 pandemic, which can be seen in the Supplementary Table 2 as following:
Supplementary Table 2: Search keywords used in regional languages
|
USA/UK/Singapore Philiphines/ Nigeria |
Spain |
Italy |
France |
Saudi Arabia |
Japan |
UAE |
South Korea |
Indonesia |
Malaysia |
Jordan |
Taiwan |
Vietnam |
|
Food bank |
Banco de alimentos |
banca del cibo |
Banque d'aliments |
food bank |
フードバンク |
Food bank |
푸드뱅크 |
bank makanan |
food bank |
بنك الطعام |
|
Nguồn cung thức phẩm |
|
Free food |
Alimento gratis |
cibo gratis |
Nourriture gratuite |
free food |
無料の食事 |
free food |
무료 식품/ 공짜 식품 |
makanan gratis |
makanan percuma |
free food |
食物 |
|
|
Free meal |
Comida gratis |
pasto gratis |
Plat Gratuit |
free meal |
無料の食事 |
free meal |
무료 음식/ 공짜 음식 |
makanan gratis |
makanan percuma |
وجبة مجانية |
餐 |
|
|
Food shortage |
Escasez de alimentos |
carenza di cibo |
Pénurie d'aliments |
نقص الغذاء |
食糧不足 |
Food shortage |
식량 부족/ 음식 부족 |
kelangkaan pangan |
kekurangan makanan |
نقص الغذاء |
糧食短缺 |
Thiếu hụt thúc ăn |
|
Food safety |
Seguridad alimentaria |
sicurezza di alimentari |
Sécurité alimentaire |
Food safety |
Food safety |
Food safety |
식품 안전 |
keamanan pangan |
keselamatan makanan |
food safety |
|
An toàn thưc phẩm |
|
Food hygiene |
Higiene de los alimentos |
igiene di alimentari |
Hygiène alimentaire |
الصحة الغذائية |
Food hygiene |
Food hygiene |
식품 위생 |
hygiene makanan |
hygiene makanan |
الصحة الغذائية |
|
Vệ sinh thực phẩm |
|
Food |
Alimento |
cibo |
Nourriture |
طعام |
Food |
طعام |
식품/ 음식 |
pangan/makanan |
makanan |
طعام |
食物 |
Thức ăn |
|
Meal |
Comida |
pasto |
Plat/ Repas |
وجبة |
Meal |
وجبة |
식사 |
makanan |
makanan |
وجبة |
餐 |
Bữa ăn |
|
Breakfast |
desayuno |
colazione |
Petit-Dejeuner |
وجبة افطار |
Breakfast |
وجبة افطار |
아침 |
sarapan |
sarapan |
Breakfast |
早餐 |
Bữa sáng |
|
alcohol |
Alcool |
alcool |
Alcohol |
الكحول |
アルコール |
alcohol |
술 |
alkohol |
alkohol |
الكحول |
酒精 |
Cồn |
|
Restaurant |
Restaurante |
ristorante |
Restaurant |
مطعم |
飲食店 |
Restaurant |
식당 |
restoran |
restoran |
مطعم |
餐廳 |
Nhà hàng |
|
Delivery |
Entrega |
consegna |
Livraison |
توصيل |
配達 |
Delivery |
배달 |
pengiriman |
delivery |
توصيل |
交貨 |
Giao hàng |
|
Food Delivery |
Entrega de comida |
domicilio |
Livraison de nourriture |
food delivery |
食品配達 |
Food Delivery |
배달 음식 |
pengiriman makanan |
Food Delivery |
Food Delivery |
外送 |
Giao hàng |
|
take away |
Comida para llevar |
porta via |
Plat à emporter |
take away |
取り除く |
take away |
포장 |
makanan bungkus |
bungkus/ bawa balik |
take away |
外帶 |
Take away |
|
Recipe |
Receta |
ricetta |
Recette |
recipe |
レシピ |
recipe |
레시피 |
resep |
resepi |
Recipe |
食譜 |
Cách nấu |
|
Cuisine |
Cocina |
cucina |
Cuisine |
cuisine |
料理 |
cuisine |
요리 |
masakan |
masakan |
Cuisine |
烹飪 |
ẩm thực |
|
Cake |
Pastel |
torta |
Gâteau |
كيك |
ケーキ |
cake |
케이크 |
cake |
kek |
كيك |
蛋糕 |
Bánh |
|
Netflix |
Netflix |
Netflix |
Netflix |
Netflix |
Netflix |
Netflix |
넷플릭스 |
netflix |
netflix |
Netflix |
Netflix |
Netflix |
|
Nintendo |
Nitendo |
Nintendo |
Nitendo |
Nintendo |
任天堂 |
Nintendo |
닌텐도 |
nintendo |
nintendo |
Nintendo |
任天堂 |
Nintendo |
|
Play station |
Play station |
Play station |
Play station |
بلاي ستيشن |
プレイステーション |
بلاي ستيشن |
플레이 스테이션 |
PS |
PS |
بلاي ستيشن |
家用遊戲機 |
Play station |
|
Cinema |
Cine |
cinema |
Cinéma |
سينما |
シネマ |
سينما |
영화관 |
cinema |
pawagam |
سينما |
電影 |
Rạp chiếu phim |
|
Hotel |
Hotel |
hotel |
Hotel |
فندق |
ホテル |
Hotel |
호텔 |
hotel |
hotel |
فندق |
飯店 |
Khách sạn |
|
Resort |
Resort |
resort |
Resort |
منتجع |
リゾート |
Resort |
리조트 |
resort |
resort |
resort |
度假村 |
Resort |
|
Park |
Parque |
parco |
Parc |
حديقة |
パーク |
Park |
공원 |
taman |
taman |
park |
公園 |
Công viẻn |
|
Plant |
Planta |
pianta |
Plante |
نبات |
工場 |
Plant |
식물 |
tanaman |
tanaman |
نبات |
植物 |
Thực vật |
|
flower |
Flor |
fiori |
Fleur |
زهرة |
花 |
flower |
꽃 |
bunga |
bunga |
زهرة |
花 |
Hoa |
|
Gym |
Gym |
Palestra |
Gym |
gym |
ジム |
Gym |
헬스장 |
gym |
gym |
Gym |
健身房 |
Gym |
|
exercise |
Ejercicio |
esercitazione |
Exercise |
ممارسة |
運動 |
exercise |
운동 |
Olahraga |
bersenam |
exercise |
運動 |
Thể dục |
|
outdoor |
Afuera |
fuori |
Dehors |
outdoor |
アウトドア |
outdoor |
야외활동 |
tempat terbuka |
luar taska |
outdoor |
戶外 |
Ngoài trời |
|
workout |
Workout |
workout |
Workout |
تجريب |
いい結果になる |
workout |
운동 |
workout |
workout |
workout |
重量訓練 |
Thể dục |
|
Yoga |
Yoga |
yoga |
Yoga |
Yoga |
ヨガ |
Yoga |
요가 |
yoga |
yoga |
Yoga |
瑜珈 |
Yoga |
|
sunbathing |
Baño de sol |
prendere il sole |
Bain de soleil |
تسفع |
日光浴 |
sunbathing |
일광욕 |
berjemur |
berjemur |
sunbathing |
日光浴 |
Tắm nắng |
|
cycling |
ciclismo |
ciclismo |
cyclisme |
cycling |
サイクリング |
cycling |
자전거 타기/ 사이클링 |
bersepeda |
berbasikal |
cycling |
騎腳踏車 |
Đạp xe đạp |
|
Vitamin |
Vitamina |
vitamina |
Vitamine |
فيتامين |
ビタミン |
Vitamin |
비타민 |
vitamin |
vitamin |
فيتامين |
維生素 |
Vitamin |
|
Vitamin C |
Vitamina C |
vitamina C |
Vitamine C |
فيتامين سي |
ビタミンC |
Vitamin C |
비타민 C |
vitamin C |
vitamin C |
فيتامين سي |
維生素 C |
Vitamin C |
|
Vitamin D |
Vitamina D |
vitamina D |
Vitamine D |
فيتامين د |
ビタミンD |
Vitamin D |
비타민 D |
Vitamin D |
Vitamin D |
فيتامين د |
維生素 D |
Vitamin D |
|
Vitamin A |
Vitamina A |
vitamina A |
Vitamine A |
فيتامين أ |
ビタミンA |
Vitamin A |
비타민 A |
Vitamin A |
Vitamin A |
فيتامين أ |
維生素 A |
Vitamin A |
|
Vitamin E |
Vitamina E |
vitamina E |
Vitamine E |
فيتامين هـ |
ビタミンE |
Vitamin E |
비타민 E |
Vitamin E |
Vitamin E |
Vitamin E |
維生素 E |
Vitamin E |
|
Vitamin B |
Vitamina B |
vitamina B |
Vitamine B |
فيتامين ب |
ビタミンB |
Vitamin B |
비타민 B |
Vitamin B |
Vitamin B |
فيتامين ب |
維生素 B |
Vitamin B |
|
Zinc |
Zin |
zinco |
Zinc |
زنك |
亜鉛 |
Zinc |
아연 |
Zink |
Zink |
zinc |
鋅 |
Kẽm |
|
Iron |
Hierro |
ferro |
Fer |
حديد |
鉄 |
Iron |
철 |
Zat besi |
Zat besi |
حديد |
鐵 |
Sắt |
|
Selenium |
Selenio |
selenio |
Sélénium |
السيلينيوم |
セレン |
Selenium |
셀레늄 |
Selenium |
selenium |
Selenium |
硒 |
Se |
|
Turmeric |
Curcúma |
curcuma |
Curcuma |
الكركم |
ターメリック |
turmeric |
강황 |
temulawak |
kunyit |
الكركم |
薑黃 |
Nghệ |
|
Garlic |
Ajo |
aglio |
Ail |
ثوم |
ニンニク |
Garlic |
마늘 |
bawang putih |
bawang putih |
ثوم |
蒜頭 |
Tỏi |
|
Ginger |
Jenjibre |
zenzero |
Gingembre |
زنجبيل |
ショウガ |
ginger |
생강 |
jahe |
halia |
زنجبيل |
薑 |
Gừng |
|
Onion |
Cebolla |
cipolla |
Oignon |
بصل |
玉ねぎ |
Onion |
양파 |
bawang bombay |
bawang merah |
بصل |
洋蔥 |
Hành |
|
Herbal |
Hierba |
erbacea |
Plantes medicinales |
Herb |
ハーブ |
Herb |
약초/ 허브 |
Herbal |
herba |
أعشاب |
草本植物 |
Thảo mộc |
|
Omega 3 |
Omega 3 |
Omega 3 |
Oméga 3 |
أوميغا 3 |
オメガ3 |
Omega 3 |
오메가 3 |
Omega 3 |
Omega 3 |
Omega 3 |
Omega-3脂肪酸 |
Omega 3 |
|
cookies |
Galletas |
biscotto |
Biscuits |
بسكويت |
クッキー |
Cookies |
과자 |
Cookies |
biskut |
بسكويت |
餅乾 |
Bánh cookie |
|
Bread |
Pan |
pane |
Pain |
خبز |
パン |
Bread |
빵 |
Roti |
roti |
Bread |
麵包 |
Bánh mì |
|
Pizza |
Pizza |
pizza |
Pizza |
بيتزا |
ピザ |
pizza |
피자 |
Pizza |
pizza |
بيتزا |
披薩 |
Pizza |
|
Chicken |
Pollo |
pollo |
Poulet |
دجاج |
チキン |
Chicken |
치킨 |
Ayam |
ayam |
دجاج |
雞肉 |
Thịt gà |
|
Herbal remedy |
Fitoterapia |
erbe |
Phytotérapie |
علاج عشبي |
薬草剤 |
Herbal remedy |
한방 치료/ 한방약 |
Jamu |
ubat herba |
علاج عشبي |
草藥 |
Dược thảo |
|
vegetables |
Legumbres |
verdure |
Légumes |
خضروات |
野菜 |
vegetables |
채소 |
Sayuran |
sayur-sayuran |
خضروات |
蔬菜 |
Rau xanh |
|
immunity |
Inmunologia |
immunita' |
Immunologie |
مناعة |
免疫 |
immunity |
면역 |
daya tahan tubuh |
immuniti |
vegetables |
免疫 |
Miễn dịch |
|
beer |
Cerveza |
birra |
Bière |
بيرة |
ビール |
beer |
맥주 |
Bir |
bir |
beer |
啤酒 |
Bia |
|
take out |
Eliminar |
take out |
Sortir |
take out |
取り出す |
take out |
포장 |
Bungkus |
bungkus/ bawa balik |
تخلص من |
外帶 |
Take away |
|
curcumin |
Cúrcumin |
curcumina |
Curcumine |
الكركم |
クルクミン |
الكركم |
커큐민 |
Kunyit |
kunyit |
الكركم |
薑黃素 |
Curcumin |
|
weigh loss |
Perdida de peso |
perdita di peso |
Perte du poids |
weigh loss |
減量 |
فقدان الوزن |
체중 감량/ 체중 감소 |
Penurunan berat badan |
penurunan/ kejatuhan berat badan |
فقدان الوزن |
減肥 |
Giảm cân |
|
plant-based diet |
dieta a base de plantas |
Dieta a base vegetale |
régime à base de plantes |
plant-based diet |
|
|
|
Diet vegan |
Plant-based diet |
|
|
Chế độ ăn thuần thực vật |
|
Ketogenic diet |
|
Dieta chetogenica |
Régime cétogène |
Ketogenic diet |
|
|
|
Diet Keto |
Ketogenic diet |
|
|
Giảm cân keto |
|
coffe |
café |
cafe' |
Café |
قهوة |
コーヒー |
coffee |
커피 |
Kopi |
kopi |
قهوة |
咖啡 |
|
|
fitness |
fitness |
Fitness |
Fitness |
fitness |
フィットネス |
fitness |
피트니스 |
Fitness |
fitness |
fitness |
weigh loss |
|
|
aerobics |
Aeróbico |
'aerobica |
Aérobique |
التمارين الرياضية |
エアロビクス |
aerobics |
에어로빅 |
Aerobik |
aerobik |
التمارين الرياضية |
coffe |
Aerobic |
|
meditation |
Meditacion |
meditazione |
Méditation |
تأمل |
瞑想 |
meditation |
명상 |
Meditasi |
meditasi |
تأمل |
Thiền |
|
|
relaxation |
Relajaxion |
rilassamento |
Relaxation |
استرخاء |
リラクゼーション |
relaxation |
휴식 |
Relaksasi |
relak |
استرخاء |
Thư giãn |
Results
Point 5. L 158 It’s not mentioned if there are differences between countries for the Diet-related lifestyle, as it appears in the rest of the categories.
Response 5: Thank you for your comment. We have added this sentence to the text. However, the top five search countries ranged across all continents from developed to developing countries (supplementary table 3). (Page 5, Line 146)
Suplementary Table 3. Top five countries in search keyword related to dietary and lifestyle behavior.
|
Keyword |
Top 1 |
Top 2 |
Top 3 |
Top 4 |
Top 5 |
|
Restaurant |
France |
Switzerland |
Germany |
Nederlands |
Spain |
|
Delivery |
Brazil |
Philippines |
Russia |
UK |
Sri Lanka |
|
Food Delivery |
Sri Lanka |
Singapore |
USA |
Malaysia |
Philippines |
|
Netflix |
Turkey |
Brazil |
Mexico |
Italy |
Spain |
|
Recipe |
New Zealand |
South Africa |
United States |
Australia |
Philippines |
|
Cake |
Indonesia |
India |
Malaysia |
Singapore |
UK |
|
Nintendo |
Japan |
Germany |
Spain |
Netherlands |
Mexico |
|
Cuisine |
France |
Belgium |
Switzerland |
Canada |
Spain |
|
Hotel |
Columbia |
Spain |
Mexico |
Argentina |
Austria |
|
Park |
United States |
South Africa |
Australia |
New Zealand |
Canada |
|
Cinema |
Brazil |
Egypt |
Italy |
Romania |
France |
|
Resort |
Russia |
Thailand |
Philippines |
Bangladesh |
Vietnam |
|
Outdoor |
Germany |
Turkey |
Switzerland |
Brazil |
Thailand |
|
Exercise |
Pakistan |
Hongkong |
Nigeria |
India |
Philippines |
|
Gym |
Mexico |
Argentina |
Spain |
France |
Sweden |
|
Plant |
Nigeria |
Netherlands |
Philippines |
Belgium |
Kenya |
Discussion
Point 6. L212 The authors refer to “For those who remain relatively healthy,..” but I understand that the health status of those who have carried out the searches is not known.
Response 6: Thank you for your suggestion. We have deleted “for those who remain relatively healthy” from the text (Page 8, L189).
Point 7. In general, it is surprising that the authors did not use search terms related to food, food patterns or type of diets (plant-based, weight loss diets, ...) distribution of meals, in addition to food delivery, shopping online, etc. They mention (L250) studies where questions of this type are addressed to know changes in dietary behaviors (Aman et al).
Response 7: Thank you for your comment. To address question of changes of dietary behaviors during covid-19 pandemic, we initially search “dietary-related keywords” such as plant-based, weight loss diets, ketogenic diets; however, those keywords showed no significant trend, except in Vietnam. We found that Vietnamese had increased searched interests for “weight loss”.
Point 8. Likewise, the limitation in language, as they indicate in L281, is especially important when comparing results by countries.
Response 8: Thank you for your comment. In order to minimize the impact of “English as the major search keyword”, we include native speakers from multi-countries who translate the “English keyword” to their mother tongue, which can be seen in Supplementary Table 2. Using the regional languages, we found a similar global search interests during the early outbreak of COVID-19.
Bibliography
Point 9. Check the correct citation of reference 6.
Response 9: Thank you for your comment. We have checked the reference 6 which is correct. This paper was written by a group of international co-authors so the authorship appears to be enormous (Page 10, Line 313).

Reviewer 2 Report
Interesting article – wondered why they did not report in abstract/summaries more on vitamin D.
In the introduction – first paragraph was repeated beginning on line 74
Was not familiar with the term “infodemiology”
Figures – found somewhat confusing. What are the listings for example for Figure 1 C, D, E, and F. -- is one column the highest search and then the lower case listing the search terms?? Needs to be better described.
Philippines with and s at the end
Consider putting the title of the Figures at the top (but may not be the journal style)…..confusing what you are looking at – have to search for the title. Maybe use Figure 1A, Figure 1B instead of just A, B
Section 3.5 has an indented listing of the Spearman’s correlations…font changed, is this a footnote?? Should it be just another paragraph?
Discussion:
How can they make the statement: “For those who remain relatively healthy??”- they were making comparisons of Global trends of covid so how do they know the “healthy” people were concerns and doing internet searches??
Discussion talks about Vitamin D but does not report correlation (r) in the listing of correlations? Some evidence that vitamin D may protect against covid .

Author Response
Reviewer 2 comments:
Point 1. Interesting article – wondered why they did not report in abstract/summaries more on vitamin D.
Response 1: Thank you for your positive response to our work. We have put correlation results of vitamin D in the abstract (Page 1, Line 42).
Point 2. the introduction – first paragraph was repeated beginning on line 74
Response 2: Sorry for our careless mistake. We have deleted the repeated paragraphs in the introduction section (Page 2, Line 75-100)
Point 3. Was not familiar with the term “infodemiology”
Response 3: Thank you for your response. Eysenbach et al. define Infodemiology as the science of distribution and determinants of information in an electronic medium, specifically the Internet, or in a population, with the ultimate aim to inform public health and public policy. Infodemiology data can be collected and analyzed in near real time. Examples for infodemiology applications include: the analysis of queries from Internet search engines to predict disease outbreaks (e.g. influenza); monitoring peoples' status updates on microblogs such as Twitter for syndromic surveillance; detecting and quantifying disparities in health information availability; identifying and monitoring of public health relevant publications on the Internet (e.g. anti-vaccination sites, but also news articles or expert-curated outbreak reports); automated tools to measure information diffusion and knowledge translation, and tracking the effectiveness of health marketing campaigns [1].
Referencee :
- Eysenbach, G.J.J.o.m.I.r. Infodemiology and infoveillance: framework for an emerging set of public health informatics methods to analyze search, communication and publication behavior on the Internet. 2009, 11, e11.
Point 4. Figures – found somewhat confusing. What are the listings for example for Figure 1 C, D, E, and F. -- is one column the highest search and then the lower case listing the search terms?? Needs to be better described.
Response 4: Thank you for your suggestion. We have modified figure legend of Figure 1 as (A) Global trend curves for coronavirus-related search terms, cumulative confirmed coronavirus cases, (B) food insecurity RSV curves. (C-F) indicates food insecurity interest heatmap covering from 31 December 2019 to 25 April 2020: (C) Food bank, (D) Free food, (E) Free meal, (F) Food shortage. (C-F) the left line indicates the top five searched countries and the right line indicated the top five rising queries related search term. (Page 4, Line 134-138).
Point 5. Philippines with and s at the end
Response 5: Thank you, we have changed “Philippine” to “Philippines” in figure 1 (Page 4, Figure 1.E and 1.F).
Point 6. Consider putting the title of the Figures at the top (but may not be the journal style), confusing what you are looking at – have to search for the title. Maybe use Figure 1A, Figure 1B instead of just A, B
Response 6: Thank you for your suggestion. We have changed all the figures as figure (1A, 1B, 1C,..(Page 4, Line 133), figure 2A, 2B, 2C… (Page 5, Line 157), figure 3A, 3B, 3C..(Page 6, Line 160) according to your suggestion.
Point 7. Section 3.5 has an indented listing of the Spearman’s correlations…font changed, is this a footnote?? Should it be just another paragraph?
Response 7: Thank you for your comment. We have put the text after the table 1 as below : (Page 7, Line 177)
Search query |
Spearman's correlation |
|
|||||
|
COVID-19 daily confirmed cases |
COVID-19 cumulative confirmed cases |
"Coronavirus" Google Trend search volume |
|||||
|
r |
p value |
r |
p value |
r |
p value |
|
|
|
Vitamins |
0.752 |
<0.001 |
0.800 |
<0.001 |
0.913 |
<0.001 |
|
|
Vitamin A |
0.780 |
<0.001 |
0.813 |
<0.001 |
0.741 |
<0.001 |
|
|
Vitamin B |
0.707 |
<0.001 |
0.745 |
<0.001 |
0.644 |
<0.001 |
|
|
Vitamin C |
0.802 |
<0.001 |
0.827 |
<0.001 |
0.957 |
<0.001 |
|
|
Vitamin D |
0.549 |
<0.001 |
0.594 |
<0.001 |
0.705 |
<0.001 |
|
|
…….. |
|
|
|
|
|
|
|
“Spearman’s correlation analysis showed that the strongest variables in each keyword category were (1) food security (food shortage: r=0.749, food bank: r=0.660, and free food: r=0.555; all p<0.001), (2) dietary behaviors (delivery: r=0.780, restaurant: r=-0.731, take away, r=0.731, and food delivery: r=0.693; all p<0.001), (3) outdoor-related behaviors (resort: r=-0.922, hotel: r=-0.913, cinema: r=-0.844, park: r=-0.827, fitness: r=-0.817, gym: r=-0.811; plant: r=0.749, sunbathing: r=0.668, and online: r=0.670; all p<0.001), and (4) immune-related nutrients/herbs/foods (vitamin C: r=0.802, vitamin A: r=0.780, zinc: r=0.781, immune: r=0.739, vitamin E: r=0.707, garlic: r=0.667, omega-3 fatty acid: r=-0.633, Vitamin D: r=0.549, and turmeric: r=0.545; all p<0.001) (Table 1: COVID-19 confirmed cases). Similar correlation patterns were observed for COVID-19 cumulative cases or “coronavirus RSV” (Table 1)”
Point 8. Discussion: How can they make the statement: “For those who remain relatively healthy??”- they were making comparisons of Global trends of Covid so how do they know the “healthy” people were concerns and doing internet searches??
Response 8: Thank you for your suggestion. We have removed “for those who remain relatively healthy” from the text (Page 8, Line 189).
Point 9. Discussion talks about Vitamin D but does not report correlation (r) in the listing of correlations? Some evidence that vitamin D may protect against covid-19.
Response 9: Thank you for your suggestion. We have reported vitamin D in Table 1 (page7, Line 177).
Search query |
Spearman's correlation |
|
|||||
|
COVID-19 daily confirmed cases |
COVID-19 cumulative confirmed cases |
"Coronavirus" Google Trend search volume |
|||||
|
r |
p value |
r |
p value |
r |
p value |
|
|
|
Vitamins |
0.752 |
<0.001 |
0.800 |
<0.001 |
0.913 |
<0.001 |
|
|
Vitamin A |
0.780 |
<0.001 |
0.813 |
<0.001 |
0.741 |
<0.001 |
|
|
Vitamin B |
0.707 |
<0.001 |
0.745 |
<0.001 |
0.644 |
<0.001 |
|
|
Vitamin C |
0.802 |
<0.001 |
0.827 |
<0.001 |
0.957 |
<0.001 |
|
|
Vitamin D |
0.549 |
<0.001 |
0.594 |
<0.001 |
0.705 |
<0.001 |
|
|
…….. |
|
|
|
|
|
|
|
We also mention it in discussion as: “Spearman's correlation analysis showed strong to moderate correlations between COVID-19 confirmed cases and vitamin C (r=0.802), vitamin A (r=0.780), zinc (r=0.781), vitamin E (r=0.707), and to a lesser extent, garlic (r=0.667), omega-3 fatty acid (r=-0.633), vitamin D (r=0.549), and turmeric (r=0.545)” (Page 8 ,Line 199-202).

Round 2
Reviewer 1 Report
Thank you for considering the changes I’d proposed. Still, I think that some aspects require more information or some modification (blue text)
Point 4. L124 It is necessary to better justify the keywords used in each category. Specially in those related to Dietary behavior
Response 4: Thank you for your comment. Prior to identify the included keyword in each category, we have searched several keywords in Google Trends platform to understand what is the global search interests related to changes of dietary behavior and lifestyle during COVID-19 pandemic, which can be seen in the Supplementary Table 2 as following:
Reviewer: Table 2 is not mentioned in the manuscript, please include it.
Review and correct some errors in the words in Spanish in table 2: alcool for alcohol, Nitendo for Nintendo, baño de sol for tomar el sol, zin por zinc, cúcuma for curcuma
Likewise, and related to point 7, it would be interesting to comment a little more on the process or reason for selecting keywords and include them in Methods section
Response 7: Thank you for your comment. To address question of changes of dietary behaviors during covid-19 pandemic, we initially search “dietary-related keywords” such as plant-based, weight loss diets, ketogenic diets; however, those keywords showed no significant trend, except in Vietnam. We found that Vietnamese had increased searched interests for “weight loss”.
Discussion
L240 and L241 Jezewska et al. and Santaliestra-Pasías references do not appear in the bibliography, please add them.
It would be convenient to include and discuss other studies related to changes in dietary behaviours during covid such as https://doi.org/10.3390/nu12061730or https://doi.org/10.3390/nu12061657, for example.
Bibliography
Point 9. Check the correct citation of reference 6.
Response 9: Thank you for your comment. We have checked the reference 6 which is correct. This paper was written by a group of international co-authors so the authorship appears to be enormous (Page 10, Line 313).
Reviewer: I was referring to the use of capital letters in the name of the first author, please change it.

Author Response
Point 1: Table 2 is not mentioned in the manuscript, please include it. Likewise, and related to point 7, it would be interesting to comment a little more on the process or reason for selecting keywords and include them in Methods section
Response 1: Thank you for your comment. We have mentioned supplementary table 2 in manuscript as following your suggestion. However, we changed sequences supplementary table 2 to supplementary table 1 as below:
“Prior to identify the included search terms in each category, we have searched several search terms in Google Trends platform to understand what is the global search interests related to changes of dietary behavior and lifestyle during COVID-19 pandemic which can be seen in the supplementary table 1. We selected those search terms that showed significant trend which were considered including “coronavirus”, “Covid-19”, “Covid 19”, “Covid”, and “SARS-CoV2” search terms”. (Page 3, Line 96-101).
Point 2. Point Review and correct some errors in the words in Spanish in table 2: alcool for alcohol, Nitendo for Nintendo, baño de sol for tomar el sol, zin por zinc, cúcuma for curcuma
Response 2 : Thank you for your comment. We have corrected the words list following your suggestion (supplementary table 1) (however we changed sequences supplementary table 2 to supplementary table 1). Supplementary material.
Supplementary Table 1. Keyword search term in different languages
|
Usa/uk/singapore Philippines/ Nigeria |
Spain |
Italy |
France |
Saudi arabia |
Japan |
UAE |
South korea |
Indonesia |
Malaysia |
Jordan |
Taiwan |
Vietnam |
|
Food bank |
Banco de alimentos |
Banca del cibo |
Banque d'aliments |
Food bank |
フードバンク |
Food bank |
푸드뱅크 |
Bank makanan |
Food bank |
بنك الطعام |
|
Nguồn cung thức phẩm |
|
Free food |
Alimento gratis |
Cibo gratis |
Nourriture gratuite |
Free food |
無料の食事 |
Free food |
무료 식품/ 공짜 식품 |
Makanan gratis |
Makanan percuma |
Free food |
食物 |
|
|
Free meal |
Comida gratis |
Pasto gratis |
Plat gratuit |
Free meal |
無料の食事 |
Free meal |
무료 음식/ 공짜 음식 |
Makanan gratis |
Makanan percuma |
وجبة مجانية |
餐 |
|
|
Food shortage |
Escasez de alimentos |
Carenza di cibo |
Pénurie d'aliments |
نقص الغذاء |
食糧不足 |
Food shortage |
식량 부족/ 음식 부족 |
Kelangkaan pangan |
Kekurangan makanan |
نقص الغذاء |
糧食短缺 |
Thiếu hụt thúc ăn |
|
Food safety |
Seguridad alimentaria |
Sicurezza di alimentari |
Sécurité alimentaire |
Food safety |
Food safety |
Food safety |
식품 안전 |
Keamanan pangan |
Keselamatan makanan |
Food safety |
|
An toàn thưc phẩm |
|
Food hygiene |
Higiene de los alimentos |
Igiene di alimentari |
Hygiène alimentaire |
الصحة الغذائية |
Food hygiene |
Food hygiene |
식품 위생 |
Hygiene makanan |
Hygiene makanan |
الصحة الغذائية |
|
Vệ sinh thực phẩm |
|
Food |
Alimento |
Cibo |
Nourriture |
طعام |
Food |
طعام |
식품/ 음식 |
Pangan/makanan |
Makanan |
طعام |
食物 |
Thức ăn |
|
Meal |
Comida |
Pasto |
Plat/ repas |
وجبة |
Meal |
وجبة |
식사 |
Makanan |
Makanan |
وجبة |
餐 |
Bữa ăn |
|
Breakfast |
Desayuno |
Colazione |
Petit-dejeuner |
وجبة افطار |
Breakfast |
وجبة افطار |
아침 |
Sarapan |
Sarapan |
Breakfast |
早餐 |
Bữa sáng |
|
Alcohol |
Alcohol |
Alcool |
Alcohol |
الكحول |
アルコール |
Alcohol |
술 |
Alkohol |
Alkohol |
الكحول |
酒精 |
Cồn |
|
Restaurant |
Restaurante |
Ristorante |
Restaurant |
مطعم |
飲食店 |
Restaurant |
식당 |
Restoran |
Restoran |
مطعم |
餐廳 |
Nhà hàng |
|
Delivery |
Entrega |
Consegna |
Livraison |
توصيل |
配達 |
Delivery |
배달 |
Pengiriman |
Delivery |
توصيل |
交貨 |
Giao hàng |
|
Food delivery |
Entrega de comida |
Domicilio |
Livraison de nourriture |
Food delivery |
食品配達 |
Food delivery |
배달 음식 |
Pengiriman makanan |
Food delivery |
Food delivery |
外送 |
Giao hàng |
|
Take away |
Comida para llevar |
Porta via |
Plat à emporter |
Take away |
取り除く |
Take away |
포장 |
Makanan bungkus |
Bungkus/ bawa balik |
Take away |
外帶 |
Take away |
|
Recipe |
Receta |
Ricetta |
Recette |
Recipe |
レシピ |
Recipe |
레시피 |
Resep |
Resepi |
Recipe |
食譜 |
Cách nấu |
|
Cuisine |
Cocina |
Cucina |
Cuisine |
Cuisine |
料理 |
Cuisine |
요리 |
Masakan |
Masakan |
Cuisine |
烹飪 |
Ẩm thực |
|
Cake |
Pastel |
Torta |
Gâteau |
كيك |
ケーキ |
Cake |
케이크 |
Cake |
Kek |
كيك |
蛋糕 |
Bánh |
|
Netflix |
Netflix |
Netflix |
Netflix |
Netflix |
Netflix |
Netflix |
넷플릭스 |
Netflix |
Netflix |
Netflix |
Netflix |
Netflix |
|
Nintendo |
Nintendo |
Nintendo |
Nintendo |
Nintendo |
任天堂 |
Nintendo |
닌텐도 |
Nintendo |
Nintendo |
Nintendo |
任天堂 |
Nintendo |
|
Play station |
Play station |
Play station |
Play station |
بلاي ستيشن |
プレイステーション |
بلاي ستيشن |
플레이 스테이션 |
PS |
PS |
بلاي ستيشن |
家用遊戲機 |
Play station |
|
Cinema |
Cine |
Cinema |
Cinéma |
سينما |
シネマ |
سينما |
영화관 |
Cinema |
Pawagam |
سينما |
電影 |
Rạp chiếu phim |
|
Hotel |
Hotel |
Hotel |
Hotel |
فندق |
ホテル |
Hotel |
호텔 |
Hotel |
Hotel |
فندق |
飯店 |
Khách sạn |
|
Resort |
Resort |
Resort |
Resort |
منتجع |
リゾート |
Resort |
리조트 |
Resort |
Resort |
Resort |
度假村 |
Resort |
|
Park |
Parque |
Parco |
Parc |
حديقة |
パーク |
Park |
공원 |
Taman |
Taman |
Park |
公園 |
Công viẻn |
|
Plant |
Planta |
Pianta |
Plante |
نبات |
工場 |
Plant |
식물 |
Tanaman |
Tanaman |
نبات |
植物 |
Thực vật |
|
Flower |
Flor |
Fiori |
Fleur |
زهرة |
花 |
Flower |
꽃 |
Bunga |
Bunga |
زهرة |
花 |
Hoa |
|
Gym |
Gym |
Palestra |
Gym |
Gym |
ジム |
Gym |
헬스장 |
Gym |
Gym |
Gym |
健身房 |
Gym |
|
Exercise |
Ejercicio |
Esercitazione |
Exercise |
ممارسة |
運動 |
Exercise |
운동 |
Olahraga |
Bersenam |
Exercise |
運動 |
Thể dục |
|
Outdoor |
Afuera |
Fuori |
Dehors |
Outdoor |
アウトドア |
Outdoor |
야외활동 |
Tempat terbuka |
Luar taska |
Outdoor |
戶外 |
Ngoài trời |
|
Workout |
Workout |
Workout |
Workout |
تجريب |
いい結果になる |
Workout |
운동 |
Workout |
Workout |
Workout |
重量訓練 |
Thể dục |
|
Yoga |
Yoga |
Yoga |
Yoga |
Yoga |
ヨガ |
Yoga |
요가 |
Yoga |
Yoga |
Yoga |
瑜珈 |
Yoga |
|
Sunbathing |
Tomar el sol |
Prendere il sole |
Bain de soleil |
تسفع |
日光浴 |
Sunbathing |
일광욕 |
Berjemur |
Berjemur |
Sunbathing |
日光浴 |
Tắm nắng |
|
Cycling |
Ciclismo |
Ciclismo |
Cyclisme |
Cycling |
サイクリング |
Cycling |
자전거 타기/ 사이클링 |
Bersepeda |
Berbasikal |
Cycling |
騎腳踏車 |
Đạp xe đạp |
|
Vitamin |
Vitamina |
Vitamina |
Vitamine |
فيتامين |
ビタミン |
Vitamin |
비타민 |
Vitamin |
Vitamin |
فيتامين |
維生素 |
Vitamin |
|
Vitamin C |
Vitamina C |
Vitamina C |
Vitamine C |
فيتامين سي |
ビタミンC |
Vitamin C |
비타민 C |
Vitamin C |
Vitamin C |
فيتامين سي |
維生素 C |
Vitamin C |
|
Vitamin D |
Vitamina D |
Vitamina D |
Vitamine D |
فيتامين د |
ビタミンD |
Vitamin D |
비타민 D |
Vitamin D |
Vitamin D |
فيتامين د |
維生素 D |
Vitamin D |
|
Vitamin A |
Vitamina A |
Vitamina A |
Vitamine A |
فيتامين أ |
ビタミンA |
Vitamin A |
비타민 A |
Vitamin A |
Vitamin A |
فيتامين أ |
維生素 A |
Vitamin A |
|
Vitamin E |
Vitamina E |
Vitamina E |
Vitamine E |
فيتامين هـ |
ビタミンE |
Vitamin E |
비타민 E |
Vitamin E |
Vitamin E |
Vitamin E |
維生素 E |
Vitamin E |
|
Vitamin B |
Vitamina B |
Vitamina B |
Vitamine B |
فيتامين ب |
ビタミンB |
Vitamin B |
비타민 B |
Vitamin B |
Vitamin B |
فيتامين ب |
維生素 B |
Vitamin B |
|
Zinc |
Zinc |
Zinco |
Zinc |
زنك |
亜鉛 |
Zinc |
아연 |
Zink |
Zink |
Zinc |
鋅 |
Kẽm |
|
Iron |
Hierro |
Ferro |
Fer |
حديد |
鉄 |
Iron |
철 |
Zat besi |
Zat besi |
حديد |
鐵 |
Sắt |
|
Selenium |
Selenio |
Selenio |
Sélénium |
السيلينيوم |
セレン |
Selenium |
셀레늄 |
Selenium |
Selenium |
Selenium |
硒 |
Se |
|
Turmeric |
Curcuma |
Curcuma |
Curcuma |
الكركم |
ターメリック |
Turmeric |
강황 |
Temulawak |
Kunyit |
الكركم |
薑黃 |
Nghệ |
|
Garlic |
Ajo |
Aglio |
Ail |
ثوم |
ニンニク |
Garlic |
마늘 |
Bawang putih |
Bawang putih |
ثوم |
蒜頭 |
Tỏi |
|
Ginger |
Jenjibre |
Zenzero |
Gingembre |
زنجبيل |
ショウガ |
Ginger |
생강 |
Jahe |
Halia |
زنجبيل |
薑 |
Gừng |
|
Onion |
Cebolla |
Cipolla |
Oignon |
بصل |
玉ねぎ |
Onion |
양파 |
Bawang bombay |
Bawang merah |
بصل |
洋蔥 |
Hành |
|
Herbal |
Hierba |
Erbacea |
Plantes medicinales |
Herb |
ハーブ |
Herb |
약초/ 허브 |
Herbal |
Herba |
أعشاب |
草本植物 |
Thảo mộc |
|
Omega 3 |
Omega 3 |
Omega 3 |
Oméga 3 |
أوميغا 3 |
オメガ3 |
Omega 3 |
오메가 3 |
Omega 3 |
Omega 3 |
Omega 3 |
Omega-3脂肪酸 |
Omega 3 |
|
Cookies |
Galletas |
Biscotto |
Biscuits |
بسكويت |
クッキー |
Cookies |
과자 |
Cookies |
Biskut |
بسكويت |
餅乾 |
Bánh cookie |
|
Bread |
Pan |
Pane |
Pain |
خبز |
パン |
Bread |
빵 |
Roti |
Roti |
Bread |
麵包 |
Bánh mì |
|
Pizza |
Pizza |
Pizza |
Pizza |
بيتزا |
ピザ |
Pizza |
피자 |
Pizza |
Pizza |
بيتزا |
披薩 |
Pizza |
|
Chicken |
Pollo |
Pollo |
Poulet |
دجاج |
チキン |
Chicken |
치킨 |
Ayam |
Ayam |
دجاج |
雞肉 |
Thịt gà |
|
Herbal remedy |
Fitoterapia |
Erbe |
Phytotérapie |
علاج عشبي |
薬草剤 |
Herbal remedy |
한방 치료/ 한방약 |
Jamu |
Ubat herba |
علاج عشبي |
草藥 |
Dược thảo |
|
Vegetables |
Legumbres |
Verdure |
Légumes |
خضروات |
野菜 |
Vegetables |
채소 |
Sayuran |
Sayur-sayuran |
خضروات |
蔬菜 |
Rau xanh |
|
Immunity |
Inmunologia |
Immunita' |
Immunologie |
مناعة |
免疫 |
Immunity |
면역 |
Daya tahan tubuh |
Immuniti |
Vegetables |
免疫 |
Miễn dịch |
|
Beer |
Cerveza |
Birra |
Bière |
بيرة |
ビール |
Beer |
맥주 |
Bir |
Bir |
Beer |
啤酒 |
Bia |
|
Take out |
Eliminar |
Take out |
Sortir |
Take out |
取り出す |
Take out |
포장 |
Bungkus |
Bungkus/ bawa balik |
تخلص من |
外帶 |
Take away |
|
Curcumin |
Cúrcumin |
Curcumina |
Curcumine |
الكركم |
クルクミン |
الكركم |
커큐민 |
Kunyit |
Kunyit |
الكركم |
薑黃素 |
Curcumin |
|
Weigh loss |
Perdida de peso |
Perdita di peso |
Perte du poids |
Weigh loss |
減量 |
فقدان الوزن |
체중 감량/ 체중 감소 |
Penurunan berat badan |
Penurunan/ kejatuhan berat badan |
فقدان الوزن |
減肥 |
Giảm cân |
|
Plant-based diet |
Dieta a base de plantas |
Dieta a base vegetale |
Régime à base de plantes |
Plant-based diet |
|
النظام الغذائي النباتي |
|
Diet vegan |
Plant-based diet |
النظام الغذائي النباتي |
|
Chế độ ăn thuần thực vật |
|
Ketogenic diet |
|
Dieta chetogenica |
Régime cétogène |
Ketogenic diet |
|
|
|
Diet keto |
Ketogenic diet |
|
|
Giảm cân keto |
|
Coffe |
Café |
Cafe' |
Café |
قهوة |
コーヒー |
Coffee |
커피 |
Kopi |
Kopi |
قهوة |
咖啡 |
|
|
Fitness |
Fitness |
Fitness |
Fitness |
Fitness |
フィットネス |
Fitness |
피트니스 |
Fitness |
Fitness |
Fitness |
Weigh loss |
|
|
Aerobics |
Aeróbico |
'Aerobica |
Aérobique |
التمارين الرياضية |
エアロビクス |
Aerobics |
에어로빅 |
Aerobik |
Aerobik |
التمارين الرياضية |
Coffe |
Aerobic |
|
Meditation |
Meditacion |
Meditazione |
Méditation |
تأمل |
瞑想 |
Meditation |
명상 |
Meditasi |
Meditasi |
تأمل |
Thiền |
|
|
Relaxation |
Relajaxion |
Rilassamento |
Relaxation |
استرخاء |
リラクゼーション |
Relaxation |
휴식 |
Relaksasi |
Relak |
استرخاء |
Thư giãn |
Discussion
Point 3. L240 and L241 Jezewska et al. and Santaliestra-Pasías references do not appear in the bibliography, please add them.
Response 3: Thank you for your comment. We have added two references for line 240 and line 241 sentences as below:
“For example, Jezewska-Zychowicz et al. showed that “meat and meat products” were positively associated with television viewing [27]. Santaliestra-Pasías and colleagues also reported that television viewing or using the internet for recreational reasons was positively associated with “snacking behavior” but negatively correlated with “plant-based” and “breakfast” dietary patterns in European adolescents [28]”. (Page 10-11, Line 252-256)
- Jezewska-Zychowicz, M.; Gębski, J.; Plichta, M.; Guzek, D.; Kosicka-Gębska, M. Diet-Related Factors, Physical Activity, and Weight Status in Polish Adults. Nutrients 2019, 11, 2532, doi:10.3390/nu11102532.
- Santaliestra-Pasías, A.M.; Mouratidou, T.; Verbestel, V.; Huybrechts, I.; Gottrand, F.; Le Donne, C.; Cuenca-García, M.; Díaz, L.E.; Kafatos, A.; Manios, Y., et al. Food Consumption and Screen-Based Sedentary Behaviors in European Adolescents: The HELENA Study. Archives of Pediatrics & Adolescent Medicine 2012, 166, 1010-1020, doi:10.1001/archpediatrics.2012.646 %J Archives of Pediatrics & Adolescent Medicine.
Point 4. It would be convenient to include and discuss other studies related to changes in dietary behaviours during covid such as https://doi.org/10.3390/nu12061730 or https://doi.org/10.3390/nu12061657, for example.
Response 4: Thank you for your comment. We have added the two references in discussion as below :
“ In addition, home confinement was also associated with unhealthy dietary patterns as participants reported increased frequencies of eating unhealthy food, eating out of control, snacking between meals, and having an increased number of meals per day [6,25]. In contrast, a study of dietary behaviors of the Spanish adult population resulted outlined healthier dietary behaviors (e.g. decreased the intake of fried foods, snacks, fast foods, red meat, pastries or sweet beverages, but increased, olive oil, vegetables, fruits or legumes) during the confinement during COVID-19 outbreak when compared to previous habits [26]”. (Page 10, Line 243-248).
- Sidor, A.; Rzymski, P. Dietary Choices and Habits during COVID-19 Lockdown: Experience from Poland. J Nutrients 2020, 12, 1657.
- Rodríguez-Pérez, C.; Molina-Montes, E.; Verardo, V.; Artacho, R.; García-Villanova, B.; Guerra-Hernández, E.J.; Ruíz-López, M.D. Changes in Dietary Behaviours during the COVID-19 Outbreak Confinement in the Spanish COVIDiet Study. J Nutrients 2020, 12, 1730.
Response:
Bibliography
Point 5: I was referring to the use of capital letters in the name of the first author, please change it.
Response 5: Thank you for your comment. Sorry for our misunderstanding. We changed the bibliography as your suggestion. (Page1, Line 327-330)
6. Ammar, A.; Brach, M.; Trabelsi, K.; Chtourou, H.; Boukhris, O.; Masmoudi, L.; Bouaziz, B.; Bentlage, E.; How, D.; Ahmed, M., et al. Effects of COVID-19 home confinement on physical activity and eating behaviour Preliminary results of the ECLB-COVID19 international online-survey. medRxiv and bioRxiv 2020, https://doi.org/10.1101/2020.05.04.20072447, doi:https://doi.org/10.1101/2020.05.04.20072447.
